# Generation of Reactive Oxygen Species by Mitochondria

**DOI:** 10.3390/antiox10030415

**Published:** 2021-03-09

**Authors:** Pablo Hernansanz-Agustín, José Antonio Enríquez

**Affiliations:** 1Fundación Centro Nacional de Investigaciones Cardiovasculares Carlos III CNIC, 28029 Madrid, Spain; 2Centro de Investigaciones Biomédica en Red de Fragilidad y Envejecimiento, Saludable-CIBERFES. Av, Monforte de Lemos, 28029 Madrid, Spain

**Keywords:** mitochondria, ROS, mechanism, signalling, disease, supercomplexes

## Abstract

Reactive oxygen species (ROS) are series of chemical products originated from one or several electron reductions of oxygen. ROS are involved in physiology and disease and can also be both cause and consequence of many biological scenarios. Mitochondria are the main source of ROS in the cell and, particularly, the enzymes in the electron transport chain are the major contributors to this phenomenon. Here, we comprehensively review the modes by which ROS are produced by mitochondria at a molecular level of detail, discuss recent advances in the field involving signalling and disease, and the involvement of supercomplexes in these mechanisms. Given the importance of mitochondrial ROS, we also provide a schematic guide aimed to help in deciphering the mechanisms involved in their production in a variety of physiological and pathological settings.

## 1. Introduction

All metazoans require oxygen to survive as it is used by the mitochondria to obtain operational energy. Mitochondria are composed by a double membrane, the inner mitochondrial membrane (IMM) and the outer mitochondrial membrane (OMM), which are separated by the intermembrane space (IMS) and hold the interior of the organelle, the mitochondrial matrix, isolated form the cytoplasm. These organelles use a panoply of carbon sources, from catabolic to anabolic pathways, which generate reducing equivalents useful for the formation of adenosine 5′-triphosphate (ATP). Reducing equivalents, such as nicotinamide adenine dinucleotide hydrogen (NADH) or flavin adenine dinucleotide dihydrogen (FADH_2_), are co-factors in multiple reactions that store electrons derived from metabolic oxidations [1]. NADH flux independently and act as substrate of mitochondrial complex I (CI), which, in turn, reduces ubiquinone to ubiquinol. On the other hand, FADH_2_ remains linked to the enzymes that are in contact with the IMM, where they interact with CoQ to recycle their FAD cofactors by reducing ubiquinone to ubiquinol. The most prominent example of FAD-dependent enzymes in mitochondria is complex II (CII). Complex III (CIII) oxidizes ubiquinol to reduce cytochrome c, which, subsequently, donates its electron to complex IV (CIV). Finally, CIV reduces oxygen to water. CI, CIII, and CIV couple the electron flux to the ejection of H^+^ across the IMM, creating a H^+^ electrochemical gradient (negative and alkaline inside), the proton motive force (Δ*p*). Δ*p* is used by a fifth complex, the ATP synthase (CV), to transfer H^+^ back to the mitochondrial matrix in an energy-releasing process, which is used to phosphorylate adenosine 5′-diphosphate (ADP) into ATP. As long as the oxygen consumption matches the phosphorylation of ADP, the respiration is coupled to ATP synthesis. Complexes from CI to CIV comprise the mitochondrial electron transport chain (mETC), which, together with CV, form the oxidative phosphorylation system (OXPHOS; Figure 1) [1].

Δ*p* is actively generated by the OXPHOS system; therefore, it reflects an active process of charge separation across the IMM, unlike the plasma membrane in which the charge separation is carried out by diffusion potential. Δ*p* can be further decomposed in mitochondrial membrane potential (Δ*Ψm*) and pH gradient (Δ*pHm*). Δ*Ψm* is created by the difference in charge distribution across the IMM, being positive in the IMS and negative in the matrix. Similarly, the distribution of H^+^ across the IMM makes the IMS acidic and the matrix alkaline. An often-forgotten integral inner membrane protein that contribute to the H^+^ gradient is the NAD(P) transhydrogenase or NNT. NNT couples hydride transfer of reducing the equivalent between NADH and NADP to proton translocation across the inner mitochondrial membrane at the expenses of ∆*p*, generating NADPH of the proton gradient, but can work in reverse to generate ∆*p* and NADH from NADPH [2]. In addition to its role in ATP synthesis, the Δ*p* is utilized by the mitochondria as the driving force to import proteins, metabolites and to balance the ion fluxes across the IMM. Interestingly, the movement of ions and charged molecules across the IMM impacts the Δ*p*.

One of the more relevant ions that cross the IMM is Ca^2+^. Mitochondrial Ca^2+^ uptake is carried out by the inwardly rectifying mitochondrial Ca^2+^ uniporter (MCU), a highly specific channel for Ca^2+^ [3]. Thus, the MCU passes Ca^2+^ down the electrochemical gradient without coupling ATP hydrolysis or transport other ions. Mitochondrion Ca^2+^ extrusion is primarily carried out by the mitochondrial Na^+^/Ca^2+^ exchanger (NCLX) [4,5] in an electrophoretic process that mediates the extrusion of 3Na^+^ per 1Ca^2+^. The activity of NCLX depends on the Na^+^ gradient created by the fast acting mitochondrial Na^+^/H^+^ exchanger (mNHE), which parallels Na^+^ gradient to Δ*pHm* [6]. In this way, Ca^2+^ and Na^+^ homeostasis becomes engaged to the activity of OXPHOS. Notably, despite the importance of these exchangers for mitochondrial homeostasis, their molecular identities have remained unknown until recently [5,7,8], with the exception of the mNHE whose molecular identification remains to be confirmed.

All organisms are subjected to acute changes in oxygen availability (hyperoxia and hypoxia, respectively) and both result in the production of reactive oxygen species (ROS). ROS are most commonly the product of subsequent one-electron reduction of oxygen. Thus, one electron reduction of oxygen produces superoxide anion (O_2_^•−^), which is the most common first step in all ROS-producing enzymes and a very toxic species. One-electron reduction of O_2_^•−^ produces hydrogen peroxide (H_2_O_2_), which is the best-known ROS acting as second messenger, normally due to its ability of reversibly oxidizing thiols groups on proteins. Subsequent one-electron reduction of H_2_O_2_ yields hydroxyl radical (^•^OH), which is an extremely harmful ROS that is notoriously involved in toxic reactions, such as the Fenton reaction. Mitochondrial ROS are involved in numerous physiological processes [9,10,11,12,13,14,15] and also link the progression from tissue homeostasis to disease [16,17,18,19]. Very surprisingly, the mechanisms of ROS production in live cells and tissues are poorly understood. Yet, thanks to the work on isolated mitochondria, we have a progressively deeper knowledge of how these organelles produce ROS [20,21,22,23,24].

## 2. Modes of ROS Production by Mitochondria

Classical experiments with mETC inhibitors pointed to CI and CIII as the major sources of ROS within the mitochondria and the cell. From a general point of view, potential sites of ROS production are triggered depending on the respiration substrate, membrane potential and, if present, the inhibitor used.

Under normal conditions, coupled respiration on glutamate/malate (GM) or pyruvate/malate (PM) activates the Krebs cycle enzymes 2-oxoglutarate dehydrogenase (OGDH), malate dehydrogenase (MDH), and pyruvate dehydrogenase (PDH), and maintains a low membrane potential as CV is producing ATP. OGDH, PDH, and MDH reduce NAD^+^ to NADH, which is, in turn, a substrate of CI. As electrons flow down the mETC they eventually reach CIII and CIV. In this situation, the production of ROS is low, but measurable and is commonly assigned to CI, the outer ubiquinone-binding site of CIII (CIII_o_) and OGDH [22,23,24] in the so-called forward electron transfer (FET) (Figure 2A). This mode of ROS production can be exacerbated by ubiquinone-binding site inhibitors of CI, such as rotenone or piericidin A [25,26,27,28], or by inhibitors of the inner ubiquinone-binding site of CIII (CIII_i_), such as antimicyn A [29,30]. On the other hand, to decrease this type of ROS production, especially after the application of the mentioned compounds, inhibitors of the flavin site of CI, such as diphenyleneiodonium (DPI; [31,32]), CIII_o_ blockers, such as myxothiazol, stigmatellin, or mucidin [33], and OGDH inhibitors, such as succinyl phosphonate or 3-methyl-2-oxopentanoate [24,34], can be applied. 

Succinate is the substrate of the Krebs cycle enzyme succinate dehydrogenase (SDH), also known as CII. SDH is representative of a variety of FAD dependent enzymes in the IMM that reduce ubiquinone, such as glyceraldehyde-3-phosphate dehydrogenase (G3PDH), dihydroorotate dehydrogenase (DHODH), and electron-transferring flavoprotein (ETF) [35]. When reducing potential is provided by CII, or to a lesser extent by other FAD-dependent enzymes, mitochondria over reduce ubiquinone and, under conditions of mitochondrial hyperpolarization (i.e., inactive CV), electrons are able to flow back through CI, reducing NAD^+^ into NADH and producing superoxide. This process is known as reverse electron transfer (RET) (Figure 2B). RET is the mode that produces the largest levels of ROS [36] and has been observed in both physiological [12,26] and pathophysiological situations [16,37,38]. Notably, the exact site of ROS production by RET is not yet clear as some authors propose that it occurs at the flavin site of CI (CI_N_; [36,39,40]), whereas others suggest the ubiquinone-binding site of CI (CI_Q_) plus CI_N_ [41,42] or the iron-sulphur cluster N2 [43] as the main actors. The investigations of ROS production by RET have used a variety of commonly used mitochondrial drugs to halt this mechanism: (i) CI inhibitors such as rotenone, DPI [20,39] or piericidin A [42]; (ii) OXPHOS uncouplers, which are Δ*Ψm*-disrupting molecules, such as FCCP [12,20,22,44]. As CI inhibitors can impede other modes of ROS production (see below), it is recommended to not only defining RET by its sensitivity to CI blockers (e.g., rotenone sensitivity), but also by using OXPHOS uncouplers. RET can be exacerbated by the incorporation of molecules to increase Δ*p* [20], such as the CV inhibitor oligomycin or ATP, and it has been proposed that RET can be modulated by the activity of NNT as regulator of the NADH concentration [45,46]. Recently, it has been shown that mice harbouring ND6-P25L mutation in their mitochondrial DNA [47] are unable to produce rotenone-sensitive RET, though they still display some residual FCCP sensitivity [48]. Such inability is due to the capacity of CI to enter into its deactive form in every catalytic cycle, possibly allowing the enzyme to undergo active/deactive (A/D) transition very rapidly under hypoxia [48]. In addition, residual ROS production under succinate oxidation may be caused by the CIII_o_ [23,49].

As commented above, independently of the electron source, CIII is able to produce ROS upon inhibition with specific molecules. CIII_i_ inhibitors, such as Antimycin A promote the accumulation of electrons inside CIII, which reach the CIII_o_ site, producing superoxide anion. It is important to note that CIII_o_ blockers, such as myxothiazol or stigmatellin do not induce the production of ROS as they render the complex completely oxidized. Indeed, CIII_o_ inhibitors block the production of ROS by Antimycin A. 

Under specific experimental conditions, when CI and CIII are inhibited and the concentration of succinate is low, CII can produce ROS at a significant rate (Figure 3). CII can also operate in forward and reverse modes as it can accept electrons both from succinate (forward) and ubiquinol (reverse). When working in the reverse mode (Figure 3A), CII can produce ROS sensitive to the ubiquinone-binding site inhibitor atpenin A5, and to the flavin-binding inhibitor malonate. However, when working on the forward mode (Figure 3B), CII can produce ROS that are only sensitive to monate, indicating that the site of ROS production under both modes is the flavin [49]. 

To note, ROS production triggered by oxidation of G3P can also be promoted by other sources, such as CI (RET) and, more scarcely, CIII_o_ [23,49]. In very specific experimental conditions, in which CI and CIII are inhibited, DHODH has been shown to produce ROS through CII and, to a lesser extent, by itself [50]. Moreover, OGDH, branched-chain 2-oxoacid dehydrogenase (BCODH) and pyruvate dehydrogenase (PDH) are potential sources of ROS under precise experimental conditions (Table 1) [34].

## 3. ROS in Acute Hypoxia

Hypoxia is defined as the decreased availability of oxygen in cells and tissues. Low oxygen levels trigger a series of responses that are related to many physiological and pathophysiological scenarios [53]. The counterintuitive observation that hypoxia induced the production of ROS was called the ROS paradox in hypoxia [54,55]. Not surprisingly, these observations were challenged by some authors [56,57]. The discrepancy was solved after measuring the kinetics of ROS production during hypoxia and demonstrating that superoxide production in hypoxia occurs only during the first minutes of hypoxia [27,30]. Thus, the discrepancy relayed in the variable timing at which the ROS measurements were taken [57]. Nowadays, though the relationship between chronic adaptation to hypoxia and the production of ROS is still unclear, acute responses to hypoxia is undoubtedly associated to hypoxic ROS generation [52,58,59,60,61,62]. 

The mechanism of hypoxic ROS production has been solved recently (Figure 4). During the first minutes of hypoxia, CI undergoes the active/deactive (A/D) transition, which consists of a conformational change involving ND3 and other CI subunits. ND3 rearranges to expose its Cys39 [63,64,65,66]. A/D transition is a characteristic dormant state of CI that is not able to perform its enzymatic activity and, therefore, is not able to pump H^+^ [27,67]. As CI becomes deactive in acute hypoxia, the mitochondrial matrix acidifies and the calcium phosphate precipitates in the matrix partially dissolve, liberating free Ca^2+^. The rise in matrix [Ca^2+^] activates the mitochondrial Ca^2+^/Na^+^ antiporter (NCLX), which promotes the entry of Na^+^ into the mitochondria. Na^+^ accumulates in the matrix and interacts with the carbonyl group in the phospholipids of the inner leaflet of the IMM. The interaction Na^+^:phospholipid promotes the formation of phospholipid aggregates, which, in turn, diminish the fluidity of IMM. The decrease in IMM fluidity lowers ubiquinol transfer between CII and CIII, while the transfer between CI and CIII is preserved as they are arranged into supercomplexes [68,69]. The decrease in ubiquinol transfer between CII and CIII promotes the generation of superoxide at the level of CIII_o_ due to the uncoupling of the Q cycle. This mechanism highlights the role of Na^+^ as second messenger, controlling OXPHOS and hypoxic redox signalling [52]. 

Indeed, it has been recently demonstrated that acute inhibition of NCLX ameliorates metabolic changes in failing hearts [70]. This pathway could be inhibited by pharmacological targeting of CI by rotenone or piericidin A, or by genetic ablation of CI subunits. In addition, CIII_o_ blockers, such as myxothiazol, CII inhibitors, such as dimtheylmalonate, and NCLX inhibitors, such as CGP-37157 prevent ROS production in hypoxia. Conversely, this pathway can be triggered in normoxia after application of drugs capable of eliminating the Δ*pHm*, such as monensin or nigericin. The uncoupler FCCP was unable to inhibit hypoxic ROS at low concentrations. However, concentrations higher than 1 μM increased ROS production in normoxia and exposing FCCP pre-treated cells to hypoxia did not to further increase ROS production. This observation indicates that the hypoxic ROS pathway was being activated in normoxia by high concentration of FCCP, which induced acidification of the mitochondrial matrix [27]. 

As Na^+^:phospholipid lowers IMM fluidity and ubiquinone transfer between CII and CIII, it is possible that this interaction also controls RET by CI. In this way, by decreasing IMM fluidity, mitochondrial Na^+^ would lower ubiquinone transfer from CII to CI, restraining superoxide production. Indeed, data from Elrod’s laboratory showed that overexpression of NCLX in mouse heart decreases infarct size and ROS production in ischemia/reperfusion [71], a well-established model of RET in pathophysiology [16]. Thus, increasing mitochondrial Na^+^ would not only increase ROS by CIII_o_, but also decrease ROS by CI-RET. This poises mitochondrial Na^+^ as a modulator of mitochondrial redox signals by selecting the kind of mechanism to be triggered, the location of superoxide production and respiratory complex involved. This may be of particular importance as the downstream adaptative signals triggered by each mechanism could be different, determining the cell fate as shown in flies [38].

It remains to be understood why in isolated mitochondria ROS levels are proportional to the environmental oxygen concentration [72,73,74]. In the past, this summed up to the arguments against the defenders of the ROS paradox [22]. The fact that the production of ROS does not increase and, rather, decreases with lower oxygen concentration in experiments with isolated mitochondria, whereas it increases in cells and tissues where cellular and molecular integrity are better preserved, provides evidence that key elements are missing or disrupted in experiments with isolated mitochondria. Such discrepancy may be due to several factors. First, it may be that the procedure of extracting mitochondria disrupts the native conformation of the oxygen sensor or the integrity of the oxygen sensing machinery (i.e., calcium phosphate precipitates may partially dissolve in the extraction procedure). Second, it is also possible that the reaction buffers used with isolated mitochondria do not resemble the cytosolic composition. In this respect, routinely used respiration buffers do not contain neither Na^+^ nor Ca^2+^, which are essential components for hypoxic induced ROS production (e.g., extracellular Ca^2+^ was shown to be essential in order to increase the magnitude of ROS production), and normally do not use a mixture of substrates (e.g., glutamate, malate plus succinate), which are also involved. Third, as the molecular identity of the oxygen sensor is not known, it is possible that it would be an extramitochondrial component that becomes lost in the mitochondrial isolation procedure.

In summary, though there is still a need to clarify why isolated mitochondria do not increase ROS upon hypoxia, it is now clear that low oxygen levels increase the production of ROS in cells and tissues. In particular, it involves the management of mitochondrial Na^+^ through an unexpected interaction with phospholipids, which is, in turn, potentially implied in all those scenarios in which the hypoxic response, mitochondrial Ca^2+^ handling, and/or ROS are involved. In this aspect, finding proteins able to control IMM fluidity may be also of great importance as they may become potential targets to lower mitochondrial ROS production and control supercomplex-independent respiration.

## 4. The Production of ROS and Supercomplexes

As mentioned above, electron transfer between CI and CIII was less sensitive to Na^+^:phospholipid interaction than electron transfer between CII and CIII. Given that CI+CIII activity reflects mostly the activity of supercomplex I+III_2_ and that CII is not superassembled with CIII at significant proportions [1], it becomes clear that supercomplexes are not influenced by Na^+^ signalling [27]. This feature may be important to prevent a potentially harmful decrease in respiratory activity at the time a redox signal is delivered. On the other hand, the more plausible molecular mechanism by which this happens would be the existence of a functionally partial segmentation of the CoQ pool [1,69,75].

The assembly of CI, CIII, and CIV in supercomplexes confers novel properties to the electron transport chain. Thus, it allows the partial functional segmentation of the CoQ and cyt c pools [69,75], stabilize CI and regulate its degradation [26,76,77,78]. Though the role in CoQ and cyt c functional segmentation was initially questioned [79,80,81,82], recent kinetic and structural data have confirmed it [68,83,84,85]. 

The existence of differentiated CoQ and cyt c redox kinetics between superassembled and free complexes explain the concept of functional pools generated by the formation of the supercomplexes. This has immediate consequences on the production of ROS. In a pioneering work, CI was maintained in supercomplex forming I + III_2_ or separated into their individual complexes (I and III_2_) with detergents. These experiments were complemented by generation of liposomes reconstituted with purified CI and CIII with variable phospholipid:protein ratio to either maintain them separated or to allow the formation of SC I + III_2_ [86]. Then, the production of ROS by CI was assessed in both experimental set ups. In either model, the authors found that superassembly of respiratory complexes reduced the production of ROS [86]. This conclusion was corroborated in vivo comparing neurons, where mitochondrial complex I is predominantly assembled into supercomplexes, with astrocytes, that maintain a higher proportion of free complex I. Thus, astrocyte ROS production is several folds higher than that of neurons [87]. 

The interaction of complexes III and IV to form the N-respirasome (I + III_2_ + IV_1–2_) or the Q-respirasome (III_2_ + IV_1–2_) was also proposed to minimize ROS production from indirect observations in vivo in zebrafish [83] and in a murine model of heteroplasmy [88], indicating that the expression of supercomplex assembly factor 1 (SCAF-1; [69,83,89]) reduced ROS levels. A direct demonstration of the impact of the tight interaction between complexes III and IV in minimizing ROS production was recently provided by the analysis of partially purified N-respirasomes harbouring or lacking SCAF1 [68]. The presence of SCAF1 determine the tight interaction of CIV and CIII within the N-respirasome, while in its absence CIII and CIV are independently attached to CI [68]. Thus, SCAF1 deficient N-respirasomes were one order of magnitude less efficient in NADH oxidation and in NADH-dependent oxygen consumption, while they generate significantly more ROS [68]. Therefore, superassembly increases the efficiency of respiration while minimizing ROS production.

Having stablished the impact of superassembly in ROS production, several mechanistic questions remain to be answered. For instance, it is yet unclear whether the partial segmentation of the CoQ and cyt c pools is: (1) just a consequence of the proximity of the CI and CIII, and CIII and CIV catalytic sites; (2) or if it is a combination of this proximity and the specific composition of protein and/or phospholipids in the supercomplex microenvironment, which may induce a different partitioning of the substrate carriers between the supercomplex phospholipid environment and that of the IMM. In addition, it remains unexplored whether superassembly impacts on different mechanisms of ROS production, such as RET. Nevertheless, factors influencing superassembly could arise as potential targets for diseases in which ROS are involved.

## 5. ROS, Acute Oxygen Sensing, and Calcium Homeostasis

Sudden exposure to hypoxia promotes organismal adaptative responses, such as hyperventilation or systemic vasodilation which are driven by an arterial chemoreceptor, called carotid body (CB). In particular, the sensory elements in the CB are the glomus cells. Similarly, the pulmonary vasoconstriction (PV) represents other acute response to hypoxia. PV is mediated by pulmonary aortic smooth muscle cells (PASMCs). It has been proposed that a specific CIV subunit, *Cox4i2* is essential for oxygen sensing in both cell types [90,91,92]. This subunit decreases the affinity of CIV for oxygen, rendering more sensitive to variations in environmental oxygen.

Interestingly, the interpretation regarding the oxygen sensing mechanism differs in both systems. On one hand, PASMCs hypoxic mitochondrial hyperpolarization would be a prerequisite to superoxide release by CIII, which, through the over-reduction of upstream mETC components, promoted ROS production by uncoupling of the Q-cycle at the level of CIII [91,92,93]. On the other hand, ROS mediated oxygen sensing in glomus cells require CI [59]. Thus, in glomus cells oxygen sensing would be mediated by the accumulation of ubiquinol followed by RET at CI. This accumulation of ubiquinol would be caused by the reduced oxygen affinity of CIV in the presence of *Cox4i2* [58,90].

As mitochondrial hyperpolarization has been postulated as a necessary factor for hypoxic ROS production in PASMCs, further experiments with OXPHOS uncouplers, such as FCCP, measuring ROS in hypoxia would definitely corroborate its involvement (Table 1). In this regard, the use of OXPHOS uncouplers would also be a valuable tool to investigate whether RET is indeed the mechanism underlying ROS production in glomus cells (Table 1). Some other considerations may help to progress in the understanding of the precise mechanism of ROS production in acute oxygen sensing by these specialized cell types. CI activity has been shown essential for ROS production and hypoxic pulmonary vasoconstriction (HPV) in PASMCs [24,93]; thus, given that CII can also feed electrons to CIII and both are necessary for HPV [94], it is possible that CI is involved in ROS production by A/D transition. In addition, it was shown that 2, 4-dinitrophenol (DNP), an OXPHOS uncoupler, or FCCP promoted a biphasic response in HPV; whereas lower amounts had from mild(increase)-to-no effects on HPV, higher concentrations inhibited it. The former was attributed to a sustained vasoconstriction during normoxia, which mimicked the effect of HPV [94]. As mentioned above, OXPHOS uncouplers can activate Na^+^-hypoxia ROS mechanism in normoxia. Thus, it is possible that the mimicking effect of DNP is due to the normoxic activation of this pathway, which, in turn, could not be further activated in hypoxia due to the presence of the uncoupler (see above, discussion on FCCP and hypoxic ROS production). This hypothesis is supported by results showing that the steady-state contraction of HPV could be inhibited by NCLX blockade [52]. The latter may be explained on behalf of the effect of high concentrations of OXPHOS uncouplers, which can result in cell toxicity, loss of function and cell death. However, it remains to be elucidated whether this effect is due to inhibition of RET or to another ROS-unrelated effect. 

Of note, a reduced CIV turnover (i.e., CIV in the presence of *Cox4i2*) would immediately translate into decreased H^+^ pumping by the mETC, which, at the same rate of CV activity, would lead to mitochondrial depolarization (and not to hyperpolarization, as it has been consistently observed). Therefore, it is possible that either CV turnover decreases, ATP/ADP ratio augments, or there are other factors involved in Δ*Ψm* regulation during acute hypoxia in PASMCs, which have a larger impact on Δ*p* than that of *Cox4i2*-driven CIV brake. However, it has been shown that the ATP/ADP ratio actually decreases during acute hypoxia in PASMCs [95] and that high ADP inhibits hypoxic response in glomus cells [96]. Alternatively, it is possible that such hyperpolarization, given its rapid occurrence, is a consequence of the activation of the fast-acting mNHE [97], which would immediately transform the Δ*pHm* into Δ*Ψm*, enabling RET, as long as there was sufficient ubiquinol accumulation. Activation of mNHE, in turn, would drive to mitochondrial matrix acidification, which would promote liberation of free Ca^2+^, activation of NCLX, mitochondrial Na^+^ overload and ROS production. Whether the latter pathway occurs in specialized cells would depend on the sensitivity to OXPHOS uncouplers to test RET (Table 1), the steady state of the calcium phosphate precipitates, NCLX expression/activity and the presence of other factors (e.g., proteins, phospholipids, etc.) regulating IMM fluidity. Nevertheless, to test whether it occurs and is triggered in PASMCs and/or glomus cells, monensin can be applied during normoxia (Table 1), as it would activate the hypoxic pathway. It is worth to mention that in very specific conditions and as mNHE is a reversible antiporter, it is possible that it itself exerts the exit and entry pathways of Na^+^, which would, in turn, depend on the Na^+^ and H^+^ gradients; rendering Na^+^ accumulation in hypoxic mitochondria independent of NCLX.

Downstream targets of ROS during acute hypoxia include K^+^ channels and Ca^2+^ channels. Whereas in some cell types, the main source of Ca^2+^ has been shown to be the extracellular milieu, in others it has been proposed to be the sarcoplasmic reticulum (SR). Such differences in the Ca^2+^ sources during acute hypoxia probably reflect the dissimilar composition of Ca^2+^ channels dedicated to the specific functions of each cell type. Thus, from a common source of ROS, which is the mETC, several targets can be modified, regardless of whether they are topologically found in the SR or the plasma membrane, which, in turn, highlights the pleiotropy of ROS as second messengers in different cell types.

## 6. Mitochondrial ROS Production in (Patho)Physiology

Mitochondrial ROS and hypoxia have been involved in the onset and progression of many physiological and pathophysiological settings, ranging from adaptation to hypoxia [52,55,58,59,90,91,92], regulation of autophagy [96,98], immunity [99,100,101,102], differentiation [103,104] to cancer [18], cardiovascular diseases [16], or neurodegenerative diseases [17]. 

The dysregulation of mitochondrial ROS production has been associated with cancer. Thus, regulation of ERK MAPK signalling pathway by CIII-dependent ROS is required for Kras-induced anchorage-independent growth and tumorigenesis [105]. Loss of SdhB subunit of CII triggers an increase in ROS production and activation of hypoxic inducible factor (HIF)-dependent genes, promoting tumorigenesis [106]. In addition, metastatic progression, which is associated with metabolic reprogramming of the cancer cells and poor outcome, is promoted by mitochondrial ROS [107]. Thus, enhancing mitochondrial antioxidant capacity may be a rational cancer therapeutic approach. Indeed, targeting catalase to the mitochondrial matrix of tumour cells suppressed tumour progression and metastasis in invasive breast cancer in mice [108]. 

Mitochondrial ROS are related to cardiovascular diseases at different levels. CI RET-dependent ROS production has been involved in ischemia-reperfusion (I/R) injury [16,37,38]. As there are several features determining superoxide production by RET, this has allowed the determination of strategies to inhibit ROS-derived RET and injury. For instance, a therapeutic approach was attempted by clamping CI in its deactive state, and, thus, uncapable to do RET, with Mito-SNO, a nitrosylating agent capable of modifying Cys39 of ND3 subunit [109]. Similarly, a missense mutant mouse for the mtDNA encoded CI subunit ND6 in mouse to rapidly undergo A/D transition, showing a significant reduction on RET and a concomitant protection against I/R injury [48]. It has been also shown that mice lacking NCLX have increased ROS production during reperfusion and that its overexpression leads to cardio-protection [71]. The molecular pathways leading to injury in mice lacking NCLX are related to increased mitochondrial Ca^2+^ overload and, probably, to increased superoxide by RET since the main mitochondrial Ca^2+^ efflux pathway is not present and mitochondrial Na^+^ does not slow down ubiquinone transfer from CII to CI. In this way, the knowledge of the molecular mechanism governing ROS production in this pathological scenario has led to the opportunity to develop strategies for the treatment of this disease. Heart failure has been associated to increased ROS production by mitochondria as well. However, the molecular mechanism does not only involve the mETC. Increasing mitochondrial Ca^2+^ levels in failing myocytes promoted higher turnover of dehydrogenases in the Krebs cycle and production of NADH, which, through the action of NNT, is converted to NADPH, a key molecule in the buffer of oxidative stress. Importantly, lowering the levels of mitochondrial Ca^2+^ lead to decreased NADPH and higher H_2_O_2_ (not superoxide) levels [110,111]. These examples also highlight the importance of an accurate and specific detection of mitochondrial ROS and that increases in H_2_O_2_ do not necessarily arise from increases in superoxide, though they are interconnected species. 

Mitochondrial ROS have also been associated to neurological disorders. Ca^2+^ homeostasis is critical for neuronal work and homeostasis. Engagement of plasma membrane L-type Ca^2+^ channels during normal autonomous pacemaking in substantia nigra dopaminergic neurons promoted transient mitochondrial ROS production and vulnerability to injury in Parkinson’s disease (PD) development. This was accompanied by mild mitochondrial depolarization, which was, in turn, partially reversed by antagonists of mitochondrial uncoupling proteins [17]. Partial depolarization is known to affect NCLX activity [112], posing NCLX as a possible cause of ROS production in PD. Indeed, halt in NCLX activity has been associated with increased mitochondrial Ca^2+^ content and neuronal cell death in two PD-associated risk protein pathways [113,114]. As hypoxic ROS may be involved in PD progression, it would be expected that HIF-1α stabilizes during PD. PINK1 knock-out neurons showed increased levels of mitochondrial ROS and HIF-1α stabilization, which, in turn, led to a glycolytic switch [115]. It is possible that mitochondrial ROS-driven glycolytic switch reprograms neuron metabolism in such a way that the production of energy through glycolysis does not match the high energetic demands of the neuron, thus, facilitating its entry into apoptosis. Interestingly, modulation of CI has been associated with development of both PD and Alzheimer’s disease (AD; [116,117,118]). Given that CI can undergo activation and deactivation, and both are related to different modes of ROS production, it would be important to ascertain whether the different physiological states of CI are related to the pathogenesis in neurological disorders.

Mitochondrial diseases are a form of rare diseases very well defined genetically, but very difficult to treat as their symptomatic manifestations varies greatly with the individuals. It was shown that a mice model of mitochondrial disease extended survival and several other parameters upon exposure to chronic hypoxia [74]. The protection was shown to be independent to the HIF pathway and, interestingly, brain hyperoxia and ROS production, were related to brain damage [74]. 

In all the previous scenarios, the immune system may play a key role and mitochondrial ROS are also crucial for its proper function. Mitochondrial ROS production is activated by the ROS-triggered CII phosphorylation by Fgr tyrosine kinase [119]. This mechanism drives macrophage activation promoting antibacterial defense [120]. The increase in succinate oxidation causes ROS production by CI-RET and favors inflammation [12]. Very interestingly, blocking CII phosphorylation by Fgr tyrosine kinase or overexpression of catalase in macrophage mitochondrial matrix prevent high fat diet-induced inflammation and obesity [9]. These works highlight the importance of metabolic rewiring of mitochondria in macrophages, from ATP-forming to ROS-producing, to promote a pro-inflammatory state.

In all these settings, different mitochondrial ROS mechanisms may play a role. It is worth to mention that the source of ROS and the mechanism implied can affect greatly what molecular pathway becomes activated downstream. In this regard, ROS target proteins differ when CI or CIII was employed as the source of ROS (PMID: 25451644). In this regard, mitochondrial ROS induced by CI-RET, but not other sources, were able to induce an increase in lifespan in flies (PMID: 27076081). Thus, it is possible that antioxidant treatment may not be as effective as expected since it tackles both beneficial and deleterious ROS [121]. Thus, given the relevance of mitochondrial ROS in so many scenarios it is essential to understand the mechanisms by which they are produced, as it would provide valuable tools to study and treat an array of diseases. Surprisingly, however, the ways of ROS production by mitochondria in vivo are poorly understood. To date there are only a few cases in physiology and disease in which the molecular mechanism of mitochondrial ROS production is clear [12,16,52]. In this regard, the importance of succinate accumulation, Na^+^ homeostasis or supercomplex formation are a few examples of possible factors modulating ROS production by mitochondria in vivo. We propose a simplified workflow, involving a variety of drugs, to evaluate the mechanisms of ROS production, which may be implied in specific experimental settings (Figure 5).

With the Figure 5 chart we aim to make the study of ROS production by mitochondria easier. It should be noted, however, that this workflow should ideally be accompanied by the corresponding enzymatic activity of the complex involved, in order to discern whether the complex involved in ROS production is a specific target of the experimental setting or compound under study. The advantage of this approach is that it can be used both in isolated mitochondria and cell culture, as well as in tissue culture. Therefore, the investigator can choose whether the structural and functional integrity of mitochondria and cells remains closer to their native environment.

With the tools and hypotheses provided herein, we hope that the study of mitochondrial ROS production, in any (patho)physiological setting, becomes easier for the researchers interested in mitochondrial redox biology.

## 7. Concluding Remarks

Mitochondrial ROS, once considered toxic byproducts of the respiratory activity of the electron transport chain that need to be eliminated at all cost, are today considered critical signalling molecules playing a fundamental role in maintaining health. In addition, the general definition of ROS species is revealingly misleading, since different ROS, like superoxide or hydrogen peroxide, exert their specific functions in different manners. Moreover, the timing and the specific loci where ROS are generated seems to exert very different physiological impact. An explosive advance in the development of this scientific field in the near future is, therefore, expected.

## Figures and Tables

**Figure 1 antioxidants-10-00415-f001:**
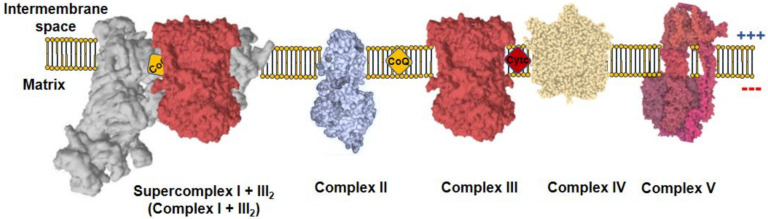
Mitochondrial oxidative phosphorylation system (OXPHOS). The inner mitochondrial membrane (IMM) comprises five protein complexes, which couple the transfer of electrons to H^+^ pumping. Charge distribution across the IMM produces a Δ*Ψm*, negative inside. Complex I (CI) is normally found inside supercomplexes with complex III (CIII) (supercomplex I + III_2_) or CIII+complex IV (CIV) (N-respirasome), a particularly relevant feature for mitochondrial reactive oxygen species (ROS) production.

**Figure 2 antioxidants-10-00415-f002:**
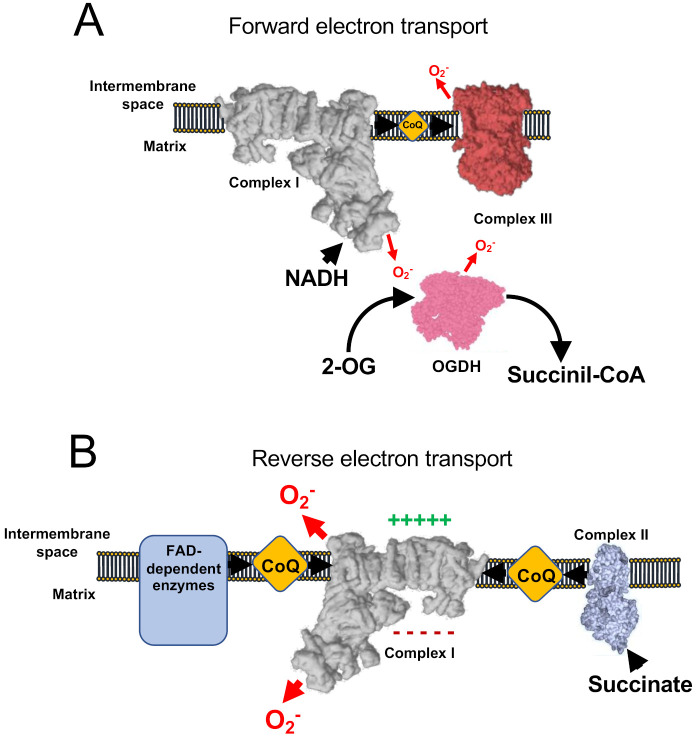
ROS production involving respiratory Complex I. (**A**) Under normal conditions, minimal amount of mitochondrial ROS are produced, nicotinamide adenine dinucleotide hydrogen (NADH) is oxidized at a high rate, as well as ubiquinol and 2-oxoglutarate. (**B**) Under conditions of normal-to-high Δ*Ψm* and accumulation of reduced CoQ (or succinate) complex I works in the reverse mode, producing high rates of ROS. In addition, a few flavin adenine dinucleotide (FAD)-dependent enzymes have also been shown to contribute to reverse electron transfer (RET).

**Figure 3 antioxidants-10-00415-f003:**
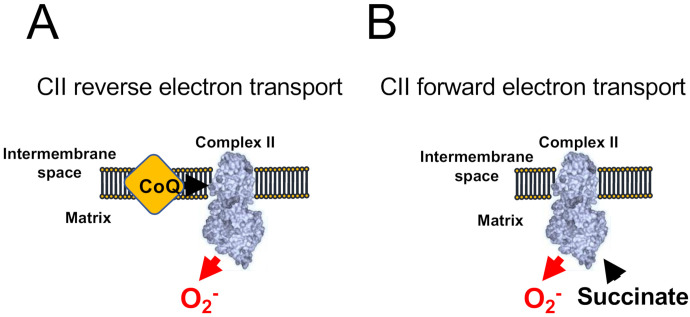
ROS production by Complex II. CII-dependent ROS production has been shown to occur only when CI and CIII are blocked. (**A**) When CoQ is highly reduced CII generates atpenin A5-sensitive ROS in its reverse reaction. (**B**) The accumulation of succinate promotes malonate-sensitive ROS in its forward reaction.

**Figure 4 antioxidants-10-00415-f004:**
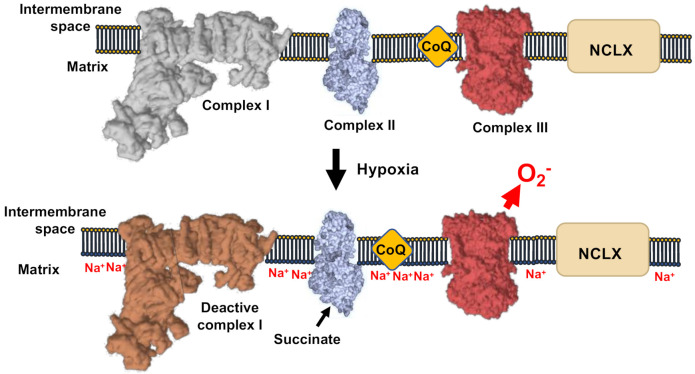
Mechanism of ROS production in hypoxia. CI-deactivation upon oxygen reduction induce de acidification of the mitochondrial matrix, the subsequent solubilization of matrix Ca^2+^ precipitates. The concomitant elevation of Ca^2+^ concentration activate the Ca^2+^/Na^+^ antiporter (NCLX). This causes the elevation of Na^+^ that decrease the inner membrane fluidity affecting the free CoQ diffusion and increase ROS production by CIII.

**Figure 5 antioxidants-10-00415-f005:**
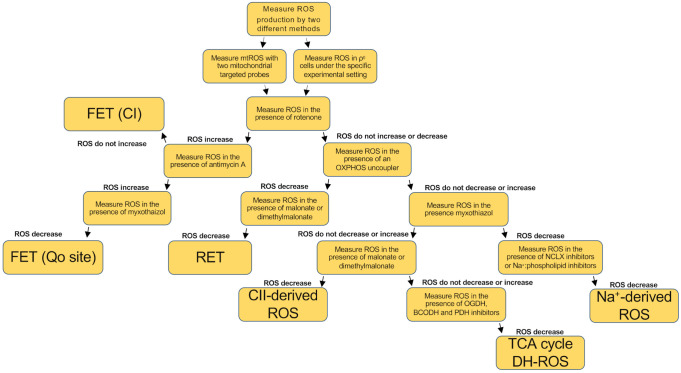
Proposed workflow to evaluate the mechanism of mitochondrial ROS production in cell culture and/or ex vivo. FET (CI) stands for forward electron transfer at the level of CI, FET (Qo site) stands for forward electron transfer at the level of Qo site, RET stands for reverse electron transport, CII-derived ROS stands for ROS production at the level of CII, TCA cycle DH-ROS stands for ROS produced by dehydrogenases at the TCA cycle and Na^+^-derived ROS stands for ROS produced by Na^+^:phospholipid interaction.

**Table 1 antioxidants-10-00415-t001:** Summary of the main ROS sources in mitochondria. Respiratory enzymes are able to produce ROS in forward and reverse reactions, as well as under specific physiological conditions, such during acute hypoxia. Biological material is underlined.

*Name and Source*	*System*	*Substrate(s)/Conditions*	*Potentiator(s)*	*Inhibitor(s)*	*References*
**FET** (**CI and CIII_o_**)	Tissues, cells, and isolated mitochondria	Cells and Tissues: standard culture media Isolated Mitochondria: pyruvate, malate and glutamate	Rotenone, Piericidin A or Antimycin A	DPI, myxothiazol, stigmatellin or mucidin	[25,26,28,29,30,31,32,33,34]
**RET** (**CI_N_**)	Cells and isolated mitochondria	Succinate or G3P	CV inhibitors or ATP	CI and CII inhibitors and OXPHOS uncouplers (FCCP)	[13,16,20,22,36,37,38,39,40,41,42,44]
**CIII_o_**	Cells and isolated mitochondria	NADH or succinate	Antimycin A	Myxothiazol or stigmatellin	[29,30,51]
**CII-derived forward ROS production**	Isolated mitochondria	Low succinate concentration, CI and CIII inhibited	-	Malonate	[49]
**CII-derived reverse ROS production**	Isolated mitochondria	Ubiquinol concentration, CI and CIII inhibited	-	Atpenin A5 and malonate	[49]
**Hypoxic ROS**	Tissues, cells and isolated mitochondria	Cells and Tissues: normal culture media Isolated Mitochondria: Malate, glutamate, CaCl_2_ and NaCl	Monensin, Nigericin, FCCP (in normoxic cells)	Rotenone, piericidin A, myxothiazol, malonate, NCLX inhibitors (preincubated)	[52]

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
