# Peer review of "Generation of Reactive Oxygen Species by Mitochondria"

_antioxidants, 2021, doi:10.3390/antiox10030415_

Round 1
Reviewer 1 Report
This review is very comprehensive and focused on the subject.
It is well written, fully intelligible, and easy to read, even for non-experts.
It is up-to-date and addresses a topic of interest to a wide audience, given the broad impact of ROS generated in the mitochondria in a multitude of pathologies. However, although many reviews address the role of mitochondria and ROS in different pathologies, this one focuses on the mechanisms and reviews important and recent contributions to these mechanisms. I have not found such detailed recent publications or reviews focused on this topic, so I think the publication of this review can be very useful and of interest to the field.
There is a detail that I think could be changed to make the abstract more accurate. Instead of saying "pathological settings", which is very general and the review does not really address pathological situations, except hypoxia, I would cite hypoxia, as a good part of the text focuses on it.
Author Response
We thank the reviewer for the helpful comments. We have added a detailed discussion on the role of mitochondrial ROS on several pathologies, including cancer, cardiovascular diseases, neurological diseases and immunity within the section “Mitochondrial ROS production in (patho)physiology”, lines 394 to 472 in the corrected version of the manuscript.
Reviewer 2 Report
The review entitled "Generation of Reactive Oxygen Species by Mitochondria" is well-written and well-organized. This review gives a nice and useful overview of the recent steps in the ROS generation inside the mitochondria, a meaningful topic being the potential cause of several diseases.
On my side, the review is suitable for publication in this format.
Author Response
We appreciate the comments of the reviewer and the constructive evaluation of the manuscript. In particular, his/her critical view regarding our discussion on the role of mitochondrial ROS on disease. In order to amend it, we have added a detailed discussion on the participation of mitochondrial ROS in a variety of pathological scenarios within the section “Mitochondrial ROS production in (patho)physiology”, lines 394 to 472 in the corrected version of the manuscript.
Reviewer 3 Report
Oxidative stress linked with an excessive reactive oxygen species generation is one of main factors leading to the development and progression of a number of diseases. Moreover ROS are involved in important physiological processes. The authors of this paper prepare comprehensive review of the current knowledge related with the molecular mechanisms of the mitochondrial ROS formation as well as its involvement in physiological and pathological conditions.
The processes related with the activity of respiratory chain complexes and mitochondrial ATP-ase, the role of mitochondrial supercomplexes in this phenomenon and effect of hypoxia are described in a great details. In particular the role of Complex II as an important source of ROS produced on FAD centre has not been widely discussed so far. Authors also indicated a number of inhibitors selective for particular complexes of Oxidative Phosphorylation which could be used to change the ROS generation rate.
The manuscript was prepared very carefully. Presented data are illustrated on 4 figures and the particular sources of ROS in mitochondria are summarized in the table. The proposed workflow (figure 4) seems to be very informative and could be helpful in the evaluation of particular mechanism of ROS formation in mitochondria. However the mitochondrial ROS related signalling in various diseases (stated in abstract and keywords) was just mentioned in the last paragraph. In my opinion it should be presented in more details.
Minor comment:
On figure 4 – the second line: “Measure ROS with two mitochondrial…” - should be “MITOCHONDRIAL”
In conclusion, the review article would be suitable for publication in Antioxidants journal after taking the above comments into account.
Author Response
We thank the reviewer for his/her positive comments.
Minor comment:
On figure 4 – the second line: “Measure ROS with two mitochondrial…” - should be “MITOCHONDRIAL”
We thanks the reviewer for noticing it. It has been corrected
Reviewer 4 Report
The manuscript is a review article associated with the mitochondria and reactive oxygen species. Comments listed below. 1. Authors may add a Figure (a graphical abstract) to introduce complex I to Complex V (ATP synthase). 2. Please also add the respiratory chain in this graphical abstract. 3. In Table 1, please add a column to cite the references.
Author Response
We thank the reviewer for his/her critical review
- Authors may add a Figure (a graphical abstract) to introduce complex I to Complex V (ATP synthase).
- Please also add the respiratory chain in this graphical abstract.
We have added a new figure (Figure 1) in which the OXPHOS and, in particular, the electron transport chain with their individual complexes are introduced.
- In Table 1, please add a column to cite the references.
We have added a new column in Table 1 listing all the references regarding each mode of mitochondrial ROS production